



# Measurement and analysis of high altitude wind profiles over the sea in a coastal zone using a scanning wind LiDAR - application to wind energy

Boris Conan [1,*] and Aleksandra Visich [2,*]

[1]Nantes Université, Centrale Nantes, CNRS, LHEEA, UMR 6598, F-44321 Nantes, France
[2]UiT The Arctic University of Norway, Postboks 385, 8505 Narvik, Norway
[*]These authors contributed equally to this work.

**Correspondence:** Boris Conan (boris.conan@ec-nantes.fr)

**Abstract.** The lack of observations at heights relevant to the wind energy industry is a major challenge for the development of the next generation offshore wind turbines expected to operate within the first tens kilometres from the coast with turbine tip reaching more than 250 m. Observations in the coastal zone, complex by its very nature as being the site of sea breezes, low-level jets, land/sea transition, are key for both understanding the marine atmospheric boundary layer processes interacting with the turbine and to the parametrization of the wind profile well above the surface layer. These needs face the difficulties of measuring in the region 150-500 m and above the sea surface. In this paper, we present an original methodology to measure the 10-min averaged wind profile at 1.5 km offshore using a scanning doppler LiDAR (Light Detection And Ranging) installed inland. The validated methodology provides a well resolved vertical profile of the horizontal wind speed and direction up to 500 m above the sea. The methodology is implemented in a 7 month test campaign in the northeastern Atlantic coast. The analysis of the wind conditions shows a proportion of low-level jets, whose origin is discussed, of 15.5% mainly coming from land and at night with a core well inside the rotor area of a 10 MW wind turbine. Wind shear events above the design values are observed 30% of the time and provide a third of the total power production. High shear events are shown to be more probable during low-level jet (68% of the time) compared to non low-level jet events (22%). The description of low-level jets and high shear events is key as they are situations where the wind profile differs from the standard values used for wind turbine design and may affect the load and fatigue predictions.

## 1  Introduction

In Europe, between 2020 and 2022, newly connected offshore wind energy capacity represents about 2.8GW per year leading to a total connected capacity of 30.3 GW. This growth is expected to be sustained to reach, in 2030, a connected capacity of 135 GW (Ramirez, 2022). For the last three years, newly installed offshore wind farms (fixed or floating) are located, on



average, in the first tens kilometres form the coast (Díaz and Soares, 2020) with an average distance to shore of 44 km in 2020, 26 km in 2021, and 43 km in 2022 (Ramirez, 2022). Even if the averaged distance to shore is tending to increase with the advent of floating offshore turbines, at that distance, the effect of the sea/land transition is still visible, therefore, from a meteorological point of view, those installations can be considered coastal rather than offshore. In 2022, 79% of the total

offshore capacity was installed in the North sea, 10% in the Irish sea, and 9% in the Baltic sea. Despite the important potential of the European Atlantic coast, analysed by its levelized cost of energy by Martinez and Iglesias (2022), the connection to the grid in 2022 of the first French offshore wind farm at Saint-Nazaire was also the first significant commercial wind farm in the entire European Atlantic coast, with 80 turbines installed at 12 km from the coast.

The growing interest of the near offshore resource led to an increasing number of studies aiming at describing the physics

of the marine atmospheric boundary layer (MABL) near the coast. This region, heterogeneous by its very nature, combines onshore and offshore complexities: orographic change (coastline), roughness change, variation of the sea-surface temperature (SST), strong contrast between ocean and land heat capacity, and enhanced dynamic interaction between water surface and the lower part of the MABL by means of waves, coastal bathymetry, local currents and tidal dynamics (Rogers, 1995; Garratt, 1994). These multi-scale interactions produce complex local atmospheric flow phenomena (Archer et al., 2014), such as the

wave boundary layer, boundary-layer transitions, coastal low-level jets (LLJ), extreme wind shear (EWS), extreme wind veer and land/sea breezes. These phenomena, in turn, cause a wide range of wind conditions and a significant deviation from the classical description of well-mixed ABL conditions (e.g. Monin-Obukhov Similarity Theory, power law). Offshore field observation, key to both an understanding of complex physical phenomena and model validation in such complex areas, is a great challenge (Hasager et al., 2008; Sempreviva et al., 2008). The advent of fixed and floating profiling LiDAR technology

allowed to make an unprecedented step in probing the MABL. However, the measurements range of a profiling LiDAR remains in the first 200-300 m of altitude that is too restrictive to understanding the above mentioned complex atmospheric flow phenomena and their interaction with wind turbines with increasing rotor diameter. The need for more offshore field data near the coastline and in a higher range of altitude remains among the current major challenges raised by Veers et al. (2019) and Shaw et al. (2022).

Older and more recent studies made from field observations (using LiDAR or metmasts) or simulations performed by Smedman et al. (1993); Mahrt (1999); Pichugina et al. (2012); Emeis (2014); Soares et al. (2014); Mahrt et al. (2014); Nunalee and Basu (2014); Pichugina et al. (2017); Wagner et al. (2019); Debnath et al. (2021); Djath et al. (2022); Aird et al. (2022); Rubio et al. (2022) and summarised in Shaw et al. (2022) point out that LLJs are quite frequent in the coastal zone. LLJs are defined by local velocity peaks in the vertical profile and high shear close to the ground with a core observed at levels between

100 m from the surface up to more than 500 m, and the phenomenon seems to be highly related to stable thermal stratification and to the coastal transition (topography, heat capacity...) Soares et al. (2014); Svensson et al. (2019). However, the physics of their formation seems multifactorial and remains unclear in many situations in the coastal environment. The complexity of modelling a LLJ near the coast with meso-scale simulation was pointed out by Nunalee and Basu (2014) and by Svensson et al. (2019) where their presence and intensity were found to be underestimated and model dependent. As LLJs can occur in

the operation altitude of offshore wind turbine, consequences are expected in the wind power generation and in the mechanical





loads on turbine structures, making the study of their characteristics of significant importance in coastal wind energy projects. To date, LLJs are studied mainly in the North sea, Baltic sea and in the US northeastern coast, while their presence in the European northwestern coast remains largely unstudied.

The expected height of next generation offshore wind turbines brings a need for measuring the wind profiles up to 400-
500 m as they will be key for the parametrization of offshore wind profile well above the surface layer. In the same time, it is a challenge that current profiling doppler LiDARs cannot tackle. In the last few years, some test campaigns deployed a scanning LiDAR (sLiDAR) to exploit its larger range in order to measure the wind speed at higher altitudes. Wagner et al. (2019) used a sLiDAR on the FINO1 platform in the North sea in DBS mode (vertical profiling) to probe up to 518 m. Although data availability was rather low, they analysed with success LLJ events within a one year period and provided tentative explanations
of their generation. Cheynet et al. (2021) used a sLiDAR with a fixed line-of-sight (LOS) at a fixed elevation angle and discussed the vertical profile of the radial wind speed (RWS) along the LOS with the limitations that the vertical profile is measured over a large horizontal distance. Shimada et al. (2020) and Shimada et al. (2022) successfully validated a method to use a sLiDAR from the shore to probe the horizontal wind speed (HWS) at a single height above the water surface using a PPI scan (Plane Position Index). These experiments are promising and efforts are still needed to measure reliable vertical profiles
of the HWS near the coast.

In the present paper, an original methodology to measure the vertical profile of the HWS above the sea is described and evaluated. Using a sLiDAR installed at the shore, it allows to measure above the sea surface up to a height of 500 m with 27 altitudes along the vertical profile. This method was implemented during a 7 month measurement campaign in the west coast of France, providing a unique data set in the European northwestern coast. Statistical analysis of LLJ and shear events is
performed to contribute to the understanding of their role in the local coastal wind resource. The paper contributes to several scientific and technical challenges identified in Veers et al. (2019) and in Shaw et al. (2022) by: (i) proposing an original field experiment method to measure the profile of the MABL up to 500m with high vertical resolution, (ii) providing a unique data set in a coastal environment in the European northeastern coast, and by (iii) contributing to the quantification of the impact of LLJs for the wind energy sector in this geographical region.

The methodology, including the original sLiDAR setup, is described in Sect. 2. Results including statistics of LLJs, wind shear events, and power production over the entire database are presented in Sect. 3. Results are discussed in Sect. 4 and a conclusion is proposed in Sect. 5. All the data treatment and visualization was performed in Python programming language.

## 2   Methodology

For this paper the *IEA 10MW 198 RW* 10 MW wind turbine is chosen for analysis with a 198 m rotor diameter and a hub height
of 119 m. (Properties were taken from https://nrel.github.io/turbine-models/Offshore.html, last access: 01 sept 2023)



**Table 1.** Configuration of the sLiDAR PPI scans.

| Scan | Azimuth $\phi$ [°] | Rot. Speed [°s$^{-1}$] | Dir. Rot. [-] | Elevation $\theta$ [°] | Acc. time [s] | Duration [s] | Distance of the gate center [m] |
|------|---------|-----------|-----------|-----------|---------|---------|----------------------------|
| PPI1 | 157.5 - 202.5 | 3 | direct | 0 | 1 | 15 | R = [100 - 3000] each 100m |
| PPI2 | 202.5 - 157.5 | 3 | indirect | 0.57 | 1 | 15 | R/(cos $\theta$) |
| PPI3 | 157.5 - 202.5 | 3 | direct | 1.72 | 1 | 15 | R/(cos $\theta$) |
| PPI4 | 202.5 - 157.5 | 3 | indirect | 4.01 | 1 | 15 | R/(cos $\theta$) |
| PPI5 | 157.5 - 202.5 | 3 | direct | 6.32 | 1 | 15 | R/(cos $\theta$) |
| PPI6 | 202.5 - 157.5 | 3 | indirect | 13.89 | 1 | 15 | R/(cos $\theta$) |

Rot. Speed: rotation speed; Dir. Rot.: direction of rotation; Acc. time: accumulation time.

## 2.1 Field experiment setup

During the period between March and September 2020, the sLiDAR (Leosphere WindCube Scanning LiDAR 100S) from the research laboratory in Hydrodynamics, Energetics and Atmospheric Environment (LHEEA) was deployed continuously on the coastline of the peninsula of Le Croisic, France (see Fig. 1), at 21 m above the mean sea level (MSL), on the balcony of a seafront villa (Fig. 2). At 100 m from the coastline, and with a clear view to the northeastern Atlantic ocean from 135° to 260°, it was set up to measure the wind resource above the ocean, some kilometres offshore from the coastline (see Fig. 1, left). A detailed description the direct environment surrounding the sLiDAR can be found in Paskin et al. (2022) who performed a field experiment in the same environment.

The hard-target procedure (Shimada et al., 2020) was used to determine the azimuth ($\phi$) angle with an uncertainty of 0.5°. Pitch and roll angles of the sLiDAR were adjusted using internal inclinometers with an uncertainty of 0.1°. For the azimuth of 180° (middle of the range used), it led to an elevation angle ($\theta$) offset of 0.07° pointing down to the water surface and corresponding to an altitude offset of -1.1 m per kilometre distance from the sLiDAR. The tide at the site is semi-diurnal with a maximal range of +/-2.7 m around the MSL. These variations were not accounted in this study, all heights are given above MSL.

The sLiDAR was configured to perform series of six 45° azimuth opening PPI scans at six elevation angles ($\theta$) above the sea surface in the southern sector (denoted PPI1-6 in Table 1). The rotation direction was alternatively direct and indirect to minimize the time between measurements. For a horizontal scan (PPI1 in Table 1) the RWS was measured at 30 gates defined at each 100 m from 1000 m to 3000 m. Range location of the other PPI scans were obtained by dividing the PPI1 ranges by "$cos\theta$" so to get measurement points on a vertical line at prescribed distances from the sLiDAR (see Fig. 3). The gate size was 50 m. All configuration details are given in Table 1. In this work, as in Gryning and Floors (2019) and in Paskin et al. (2022), the threshold carrier-to-noise ratio (CNR) value used to validate a RWS measurement was set to -29dB. RWS measured over 30 m/s are discarded as they exceed the maximum wind speed measurement given by the manufacturer.





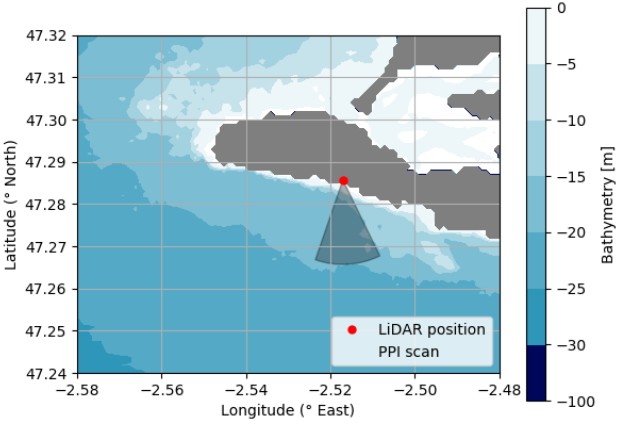
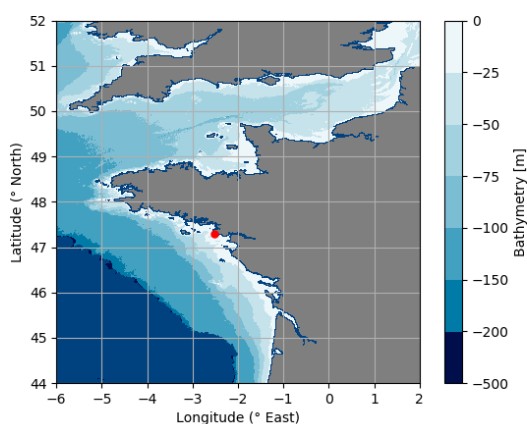

**Figure 1.** Location of the sLiDAR deployment site at Le Croisic, France (left). Position of the sLiDAR on Le Croisic peninsula (red dot) and scanned are (grey area) (right).

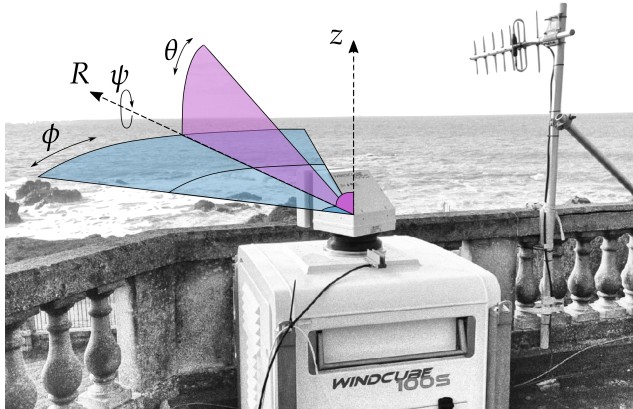

**Figure 2.** View of the sLiDAR facing the Atlantic. Definition of the sLiDAR frame with the azimuth ($\phi$) and elevation ($\theta$) angles. Picture from Paskin et al. (2022)

## 2.2 Virtual mast reconstruction approach

### 2.2.1 Horizontal wind speed estimation

Horizontal wind speed (HWS) can be defined either by its eastward and northward components ($u$, $v$), or by ($U_h$, DIR), where $U_h = \sqrt{u^2 + v^2}$ is the modulus of HWS and DIR is wind direction. The vertical component of the wind is denoted by $w$. During a scan, each RWS measured by the sLiDAR at a given gate is related to the wind speed vector components ($u$, $v$, $w$) by

$$RWS = u \cos\theta \sin\phi + v \cos\theta \cos\phi + w \cos\theta. \qquad (1)$$



Assuming that the wind field is homogeneous in space and time during a PPI scan and that $w$ is negligible compared to $(u,v)$,
Equ. 1 simplifies to

$$RWS = u \, \cos\theta \, \sin\phi \, + \, v \, \cos\theta \, \cos\phi \,, \qquad (2)$$

and the HWS components can be determined by a linear-least-square method with $\theta$, $\phi$ and $RWS$ as known values for
each PPI scan. Shimada et al. (2020) successfully validated this method (called velocity volume processing) against a profiling
LiDAR in a very similar set-up (sLiDAR at the shore measuring over the sea). They reported an accuracy in the HWS estimation
at 100m above sea level of 0.5% with a determination coefficient of 0.998. Additionally, the elevation angle was reported as
having no influence on accuracy. In the present work, a similar approach is used except that Equ. 2 was re-written as

$$RWS^* = \frac{RWS}{\cos\theta \, \sin\phi} \, = \, u \, tan\phi \, + \, v \,, \qquad (3)$$

so that the fitting is based on a linear regression procedure instead of fitting a sum of sinus functions. In this case, linear
regression will yield $u$ as the slope of the $RWS^* = f(\tan\phi)$ line and $v$ as its intercept. This approach has an obvious limitation
related to the variables in the denominator possibly close to zero. However, for the sLiDAR setting used in the measurements
(see Table 1) neither of the two functions takes near-zero values at any point. When a different range of azimuth angles $\phi$
is used, additional steps are necessary to avoid division by zero (a simple turning of the coordinate system before and after
fitting can help). Besides, it is important for a correct performance of the algorithm that the values of $\tan\phi$ are distributed
quasi-linearly across the range, i.e. that the range of azimuth angles of the sLiDAR setting lies within the linear part of the
$f(x) = \tan x$ graph (which is true in our case).

Quality of the data obtained from linear regression of Equation 3 was judged by NRMSE (Normalized Root Mean Square
Error), i.e. the standard deviation of the residuals, normalized by the maximum absolute value of $RWS^*$:

$$NRMSE = 1 - \frac{1}{max(abs(Y_i))} \sqrt{\frac{\sum_{i=1}^{N}(Y_i - \hat{Y}_i)^2}{N}} \,, \qquad (4)$$

where $Y_i$ denotes the measured value of $RWS^*$, $\hat{Y}_i$ is the value predicted by the linear regression, and $N$ is the total number
of points. Data with $NRMSE < 0.75$ are considered unreliable and discarded. After the filtering, the obtained $u$ and $v$ values
are averaged over 10-minute intervals for each measurement height.

### 2.2.2  Vertical profile reconstruction and 10-min averaging

Throughout the measurement campaign, the six PPI scans defined in Tab. 1 and displayed in Fig. 3 are performed one after
another in a loop, adding up to about 6 scans of each configuration within a 10-minutes interval, which in turn allows for
standard 10-minutes averaging once each scan has been processed separately. Each scan has pre-configured measurement
gates, i.e. distances from the sLiDAR along the scan ray at which RWS is recorded (gray circles in Fig. 3). The elevation
angles and the gates are set in such a way that at the target distance from the sLiDAR (1400 m) the Nth gates of all six scans
are vertically aligned, thereby creating a "virtual met mast" (orange circles in Fig. 3). The six measurement heights of this mast
are 0, 14, 42, 98, 155 and 346 m above the sLiDAR.





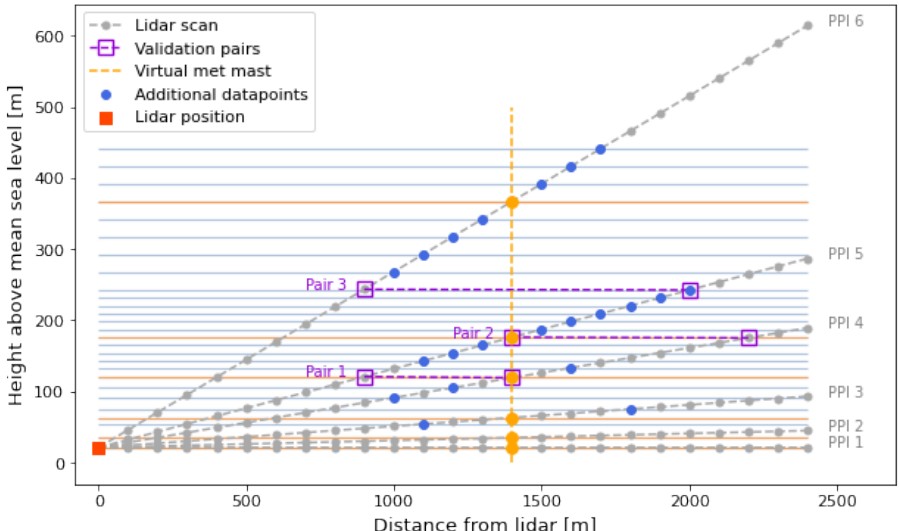

**Figure 3.** Representation of the elevation of measurement points along the PPI scans (grey dotted line). The location of the reconstructed mast is denoted in orange. Blue dots denote the additional points taken into account in the reconstruction approach. Purple squared are comparison points, at the same high, used for validation.

To ensure better height resolution of the vertical profile, RWS measurements from the gates adjacent to the ones forming the virtual met mast were included in the dataset (blue circles in Fig. 3), adding 21 new measurement heights (33, 54, 70, 84, 112, 122, 133, 144, 166, 177, 188, 199, 210, 221, 253, 276, 300, 323, 369, 393 and 416 m above the sLiDAR). The additional heights allow to extend the mast further up and provide significant grid refinement at altitudes higher than 100 m above the sLiDAR. This expansion relies on the assumption that variation of wind properties is much stronger with height than with

distance from shore within the new "width" of the mast. To validate this assumption, three validation pairs (purple in Fig. 3) were selected, where wind speed and direction were compared at the measurement points with the same height but different distance to the shore. Pair 1 and Pair 2 compare the the conditions at the boundaries of the extended mast with those at its core, while Pair 3 spans the whole extended width. The results of this comparison are shown in Fig. 4, where 10-minute averaged data from March and April 2020 were used. A more restrictive $NRMSE$ threshold of 0.96 (compared to 0.75) was applied

at the stage of wind speed estimation (see Sect. 2.2.1) in order to focus only on the method to increase height resolution method only and not other sources of uncertainty. It is to be noted that the two measurements in each pair are not performed simultaneously, so the measured wind speeds are not expected to match perfectly in a turbulent flow. However, the analysis illustrated by Fig. 4 features a remarkable correlation both in wind speed and wind direction for each of the pairs. This shows that despite the land-sea transition and ocean roughness change due to the coast, in the present case the evolution of horizontal

wind is not significant within a 1000 m horizontal range centered at the virtual met mast. Thereby the underlying assumption of the increased height resolution method is considered valid, and the method is employed for the further analysis, allowing for a more correct description of the MABL.



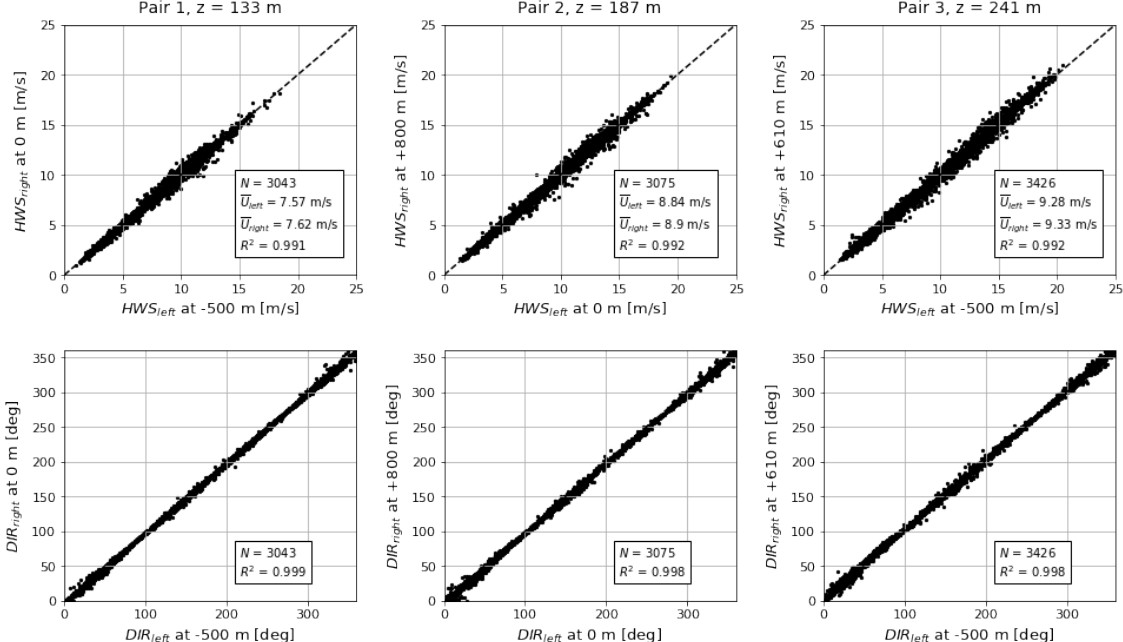

**Figure 4.** Horizontal wind speed and wind direction comparison for three validation pairs. Axis title provide the distance of each measurement point (left and right) form the virtual mast.

The obtained profiles are further subjected to reliability filtering, where a profile is considered reliable if mre than 5 valid measurements are available within the 10-minute averaging interval for at least 20 heights out of 27. The unreliable profiles are not used for further analysis.

### 2.2.3 Statistical convergence of the 10-min horizontal wind speed estimation

As it is calculated from a finite signal, $U_h$ measured at the field has a statistical convergence error of $\epsilon_{U_h} = \pm z_{\alpha/2} \frac{\sigma_u}{\sqrt{N}}$, where $z_{\alpha/2} = 1.65$ for 90% confidence level, $N$ is the number of independent samples, and $\sigma_U$ the standard deviation of $U_h$. The frequency at which the sLiDAR is computing $U_h$ is quite small, 6 times in a 10-min slot, compared to a punctual measurement such as that with an ultrasonic anemometer (USA) or a cup anemometer which is measuring the HWS at 20 Hz. However, contrary to a USA, the sLiDAR also accumulates measurements in space during each PPI so time and space contribute to increase $N$.

In time, velocity measurements are considered statistically independent when they are separated by twice the integral time scale $T_u$. In space, measurements are independent if they are spaced by twice $\Lambda_u$, the integral length scale along the wind direction (in this analysis, it is assumed that $\Lambda_u$ is homogeneous in the horizontal plane). $\Lambda_u$ can be estimated by $\Lambda_u \approx \kappa z$, and assuming Taylor frozen turbulence, (we know it is not perfect, Kaimal Finni 2.2p35) time and length scales are related by $\Lambda_u = U_h \, T_u$. Here, the maximum number of spatially independent points per scan can be approximated by $N_{sLiDAR} =$





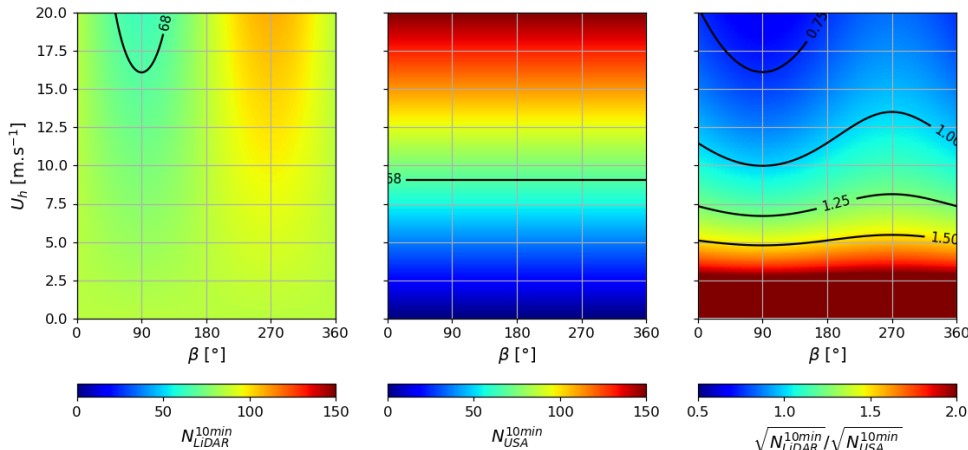

**Figure 5.** Evolution of $N_{sLiDAR}^{10min}$ (left), $N_{USA}^{10-min}$ (middle) and $R_\epsilon$(right) as function of the $U_h$ and $\beta$.

$\frac{L_{sLiDAR}}{2\Lambda_u}$ where $L_{sLiDAR} = L_{scan} - T_{scan}U_h sin\beta$ represents the equivalent length of one scan where $L_{scan} = 2Rsin\left(\frac{\Delta\theta}{2}\right)$ is the cord of a scan with $\Delta\theta$ the azimuth opening of the scan and $R$ the distance of the centre of the gate considered, and where

$T_{scan}U_h sin\beta$ is the equivalent distance travelled by the wind in the direction of the scan during the scan duration $T_{scan}$ with $\beta$ the direction of the sLiDAR beam relative to the wind ($0^o$ means the mean azimuth is aligned with the main wind direction). In the configuration tested, each scan is performed 6 times in a 10-min period leading a maximum number in a 10-min period of $N_{sLiDAR}^{10-min} = 6\frac{L_{sLiDAR}}{2\Lambda_u}$.

    In Fig. 5 a, $N_{sLiDAR}^{10-min}$ is visualized as a function of $\beta$ and $U_h$ with $z = 100m$, $\sigma_u = 1$ m/s, $R = 1500m$, and $\Delta\theta = 45°$.

$N_{sLiDAR}^{10-min}$ is not very dependent on $U_h$ and varies slightly with $\beta$, the most favourable condition is $\beta = 270°$ where the scan goes in opposite directions compared to the wind direction. The value $N_{sLiDAR}^{10-min} = 68$, denoted on the figure refers to the number of independent points necessary to reach 2% of error with 90% confidence level. As a comparison, a USA placed in the same flow will collect, in 10 minutes, $N_{USA}^{10-min} = \frac{600}{2T_u} = \frac{600U_h}{2\Lambda_u}$ independent samples with no influence of the wind direction. Figure 5b shows that $N_{USA}^{10min}$ is very sensitive to wind speed variation. Finally, the ratio of the error made by the

sLiDAR compared to a USA,

$$R_\epsilon = \frac{\epsilon_{sLiDAR}^{10min}}{\epsilon_{USA}^{10-min}} = \frac{\sqrt{N_{sLiDAR}^{10-min}}}{\sqrt{N_{USA}^{10-min}}}, \quad (5)$$

examplified in Fig. 5 c, becomes independent on $\Lambda_u$ and $\sigma_u$. For low wind speed, typically $U_h$<10 m/s, in the present configuration, the sLiDAR provides a lower statistical error compared to a USA, the opposite is observed when $U_h$>10 m/s. Note that the results would be similar when compared to a cup anemometer instead of a USA as only independent samples can be

accounted to estimate the number of independent points.



When applied to the configuration used in this work (Tab. 1), the resolution of the PPI scans (15 points in a 45° scanning sector) is sometimes lower than $N_{sLiDAR}^{10-min}$. In this case, the statistical error is estimated using $N = min(N_{sLiDAR}^{10-min}, 15)$. In the range of distance used, the statistical uncertainty with 90% confidence level varies in $\epsilon_{sLiDAR}^{10-min} = [1.74 - 4.35]\%$.

## 2.3  Definition of wind profile characteristics

### 2.3.1  Shear exponent definition

Throughout the present analysis, wind shear, also called shear exponent, $\alpha$ is defined from power law:

$$U = U_{ref} \left( \frac{Z}{Z_{ref}} \right)^{\alpha}, \tag{6}$$

where $U_{ref}$ and $Z_{ref}$ denote reference speed and reference height, respectively. This equation can be rewritten in the logarithmic form:

$$\log U = \alpha \cdot \log Z + (\log U_{ref} - \alpha \cdot \log Z_{ref}), \tag{7}$$

according to which $\alpha$ can be seen as the slope of the $U(Z)$ function in log-log scale and allows to derive the shear exponent value from several wind speed values at different heights using linear regression. The shear exponent is derived from the wind speed values within the rotor area only (i.e. between the heights closest to the bottom and the top of the reference turbine rotor), this definition was chosen to comply with wind energy related application. Note that higher shear may be present in the wind
profile inside or outside the rotor. The process was performed as part of data processing for each valid wind profile.

### 2.3.2  Low-level jet detection method

In this work, low-level jets are simply defined as a peak in the vertical wind profile. No other criteria (atmospheric stability, roughness transition) nor hypothesis on its formation, nor spatial extension is accounted for the detection. As such, it can be the result of various phenomena in the ABL. The detection is performed for every valid 10-minute averaged profile through an
algorithm based on the SciPy function `find_peaks` following three steps. First step, peaks more prominent than 1 m.s$^{-1}$ and not narrower than 5 height levels (to exclude sharp fluctuations) were detected. If the function returned more than one peak, the profile was considered as not containing a LLJ. Second step, absolute and relative low-level jet criteria were used to evaluate the discovered maximum, a method similar to the one described by Aird et al. (2022), were set to 2 m.s$^{-1}$ (absolute) and/or 20% (relative) higher than the next minimum to be considered an LLJ. Third step, the time dimension of low-level jets was
taken into account. The boolean array corresponding to the LLJ presence or absence was filtered so that an isolated 10-minute averaged profile with a detected LLJ is not accounted in the final statistics, while, on the contrary, a no-LLJ case surrounded by LLJ-cases on the timeline would be counted as such. The configuration allows to measure LLJ core down to 25 m with a 25 m vertical resolution.





# 3   Results

## 3.1   Global wind statistics

|  | Mar | Apr | May | Jun | Jul | Aug | Sep | Overall |
|---|---|---|---|---|---|---|---|---|
| **Reliable profiles [%]** | 91.8 | 92.5 | 85.8 | 94.5 | 64.0 | 60.6 | 81.0 | 81.4 |
| **Max wind speed [m.s$^{-1}$]** | 23.7 | 20.1 | 17.9 | 20.2 | 15.9 | 19.6 | 15.6 | 23.7 |
| **Mean wind speed [m.s$^{-1}$]** | 9.3 | 7.0 | 8.0 | 7.2 | 7.4 | 5.8 | 6.6 | 7.4 |
| **Maximum shear exponent** | 1.21 | 1.50 | 0.86 | 1.10 | 1.12 | 1.01 | 1.09 | 1.50 |
| **Minimum shear exponent** | -0.91 | -1.31 | -1.25 | -1.07 | -0.87 | -1.14 | -1.41 | -1.41 |
| **Mean shear exponent** | 0.14 | 0.12 | 0.14 | 0.09 | 0.10 | 0.09 | 0.16 | 0.12 |
| **LLJ occurrence [%]** | 9.0 | 13.1 | 18.8 | 8.0 | 14.8 | 16.8 | 31.3 | 15.5 |
| **Mean LLJ core speed [m.s$^{-1}$]** | 9.5 | 8.2 | 10.0 | 10.1 | 10.1 | 8.5 | 9.3 | 9.4 |
| **Mean LLJ core height [m]** | 172 | 151 | 158 | 180 | 186 | 177 | 167 | 168 |

**Table 2.** Monthly and global wind statistics from 10-min data: wind speed at hub height, shear exponent, and LLJs.

Monthly and overall wind statistics, including those on LLJs and wind shear, from the period of the test campaign are reported in Tab. 2. Data availability of the vertical profile is mostly higher than 85%. In July and August, it is below 65% mainly due to maintenance and various tests conducted in this period. The monthly mean wind speed at hub height ($U_{hub}$) varies from 5.8 m s$^{-1}$ in August to 9.3 m.s$^{-1}$ in March when the site is usually subject to a series of low-pressure cells generating storms. Figure 6a presents the 10 min averaged wind speed distribution for the entire measurement period and the fitting by a Weibull distribution expressed by

$$PDF(U_{hub}) = \frac{k}{c}\left(\frac{U_{hub}}{c}\right)^{k-1} e^{-\left(\frac{U_{hub}}{c}\right)^{k}} \tag{8}$$

where the scale parameter is $c$ and the shape parameter is $k$. These parameters are calculated from the measured data using the formulae of Justus et al. (1978).

The wind rose of 10 min wind speed at hub height, Fig. 6b, shows a clear dominance of two wind sectors: Western winds in the sector [180°-315°], resulting from Atlantic cyclonic depressions, and North-Eastern winds in the sector [0° - 90°]. This pattern is classical for this area, yet, it is should not be considered as an annual statistical representation as all the winter season and a large part of falls are not included in the dataset. Western wind may be under-represented in the dataset. Later in the text, *sea wind* and *sea wind* are used to denote winds coming from the sea [315°-135°] and from the land [135°-315°], respectively.





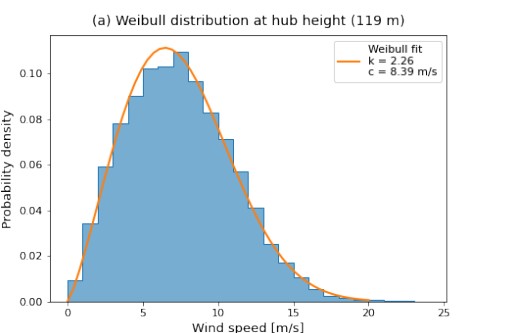
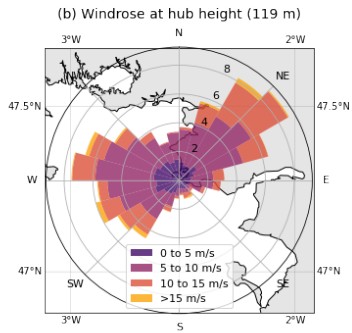

**Figure 6.** (a) Histogram of the 10-min wind speed at hub height for the entire period of measurement with a Weibull fitting (red line), (b) Wind rose of the 10-min wind direction at hub height superimposed to the local coastline with 15° resolution. Colors corresponds to hub height velocity ranges.

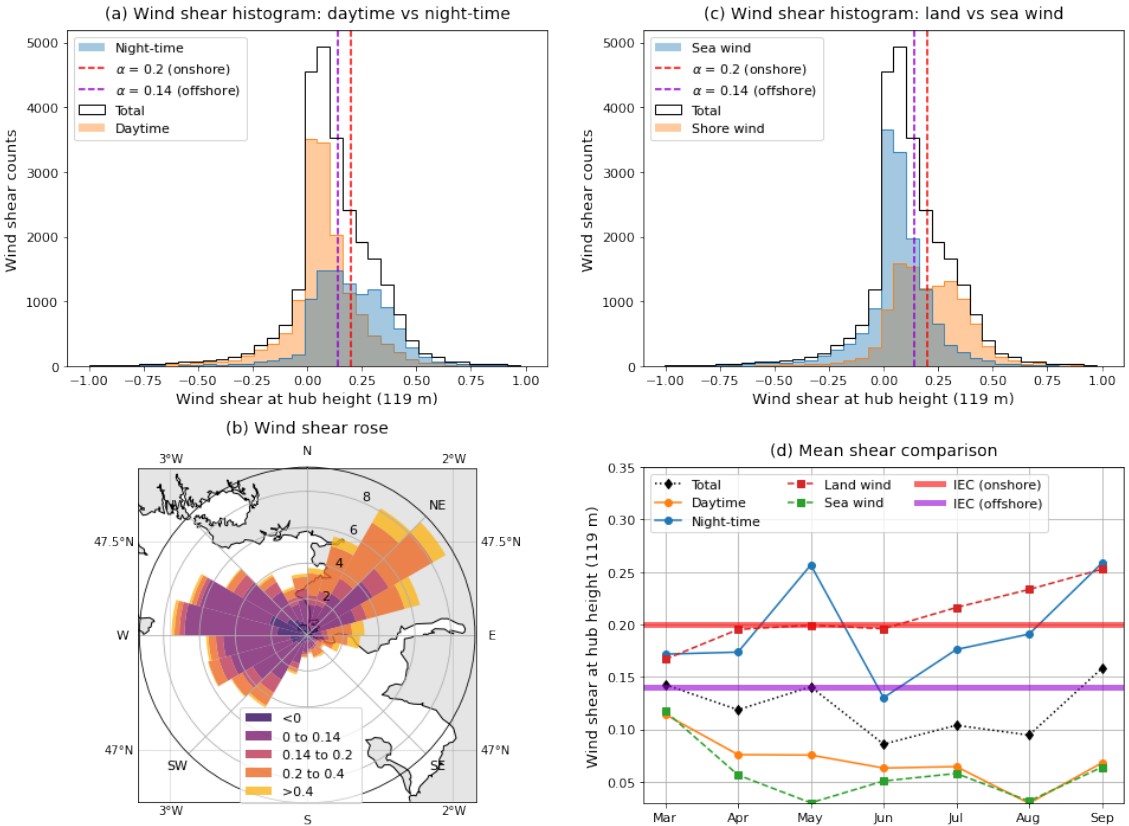

**Figure 7.** Histogram of $alpha$ comparing daytime and nighttime (a) and comparing wind directions (c). Wind rose of $\alpha$ (b) and monthly averages (d).



## 3.2 Wind shear statistics

On monthly average, $\alpha$ ranges from 0.09 to 0.16 (Tab 2). The average value of the dataset, $\alpha = 0.12$, is below the values used for offshore (0.14) and onshore (0.2) wind turbine design (IEC, 2009, 2005). However, very large variations up to levels above 1 and down to levels below -1 are also observed in the entire period.

The wind shear distribution, presented in Fig. 7a with a black contour, shows a great scatter around the mean value with a thicker tail towards higher values. The sub-distribution corresponding to *daytime* (orange in the figure) is more symmetrical with a sharp peak around a lower shear value ($\alpha \approx 0.06$), while the *nighttime* distribution (blue in the figure) is shifted towards higher values ($\alpha \approx 0.2$) with more spreading, creating the thick tail in the overall distribution. *Daytime* and *nighttime* are defined by the astronomical sunrise and sunset times. Figure 7a also illustrates that for a significant number of cases, the wind shear exponent overpasses the values given by IEC for both offshore (red dashed line) and onshore designs (purple dashed line).

In Fig. 7b, where wind shear exponents are sorted into *land wind* and *sea wind*, the *sea wind* distribution largely resembles the *daytime* distribution and the *land wind* distribution is very similar to the *nighttime* distribution. Even though a systematic link between the pairs cannot be made based only on the similarity of the histograms, a connection can be made. On the one hand, *nighttime* over land are often associated with a stable thermal stratification that leads to a higher shear exponent compared to unstable conditions often observed during *daytime* and associated with lower shear exponents. On the other hand, *sea wind*, due to lower surface roughness, is associated with a lower power exponent compared to *land wind*. Note that similar observation of the influence of thermal stability and surface roughness on the wind shear are reported in Hanafusa et al. (1986); Irwin (1979). No direct correlation is given here between *daytime/nighttime* and the wind direction.

Figure 7c presents the distribution of $\alpha$ by 15° wind direction sectors in a wind rose format. The distribution shows that *land winds* have a higher $\alpha$ compared to *sea winds*. High wind shear are clearly associated to onshore conditions meaning that some of the characteristics of inland wind persist at least up to 1.5 km from the shore.

Figure 7d presents monthly average values of $\alpha$ for the four categories (*offshore wind*, *onshore wind*, *nighttime* and *daytime*). Once again, a significant difference between the categories is observed, while daytime and offshore values lie well below the IEC offshore shear reference of 0.14, and mostly below 0.1, their *nighttime* and *land wind* counterparts are much higher and in some cases lie above the IEC onshore shear reference of 0.2.

Five ranges of $\alpha$ are given in Tab. 3 to define the wind shear: negative wind shear (NWS) representing situations where the wind speed is monotonically decreasing with height, low wind shear (LWS) below the value used in the IEC for onshore, medium wind shear (MWS), between the onshore and offshore IEC values, high wind shear (HWS) when it overpasses the IEC offshore value, and extreme wind shear (EWS) where it is more than twice the offshore IEC value. In Fig. 8, presenting the time share in each range, it is observed that the sum of LWS and MWS ranges, which are in agreement with IEC standards, are representing about 55% of the total observation time. Higher shear are observed 29% of the time (HWS and EWS), including nearly 7% of the time where it is more than twice the onshore reference (EWS). NWS events are not negligible with nearly 16% of the time.

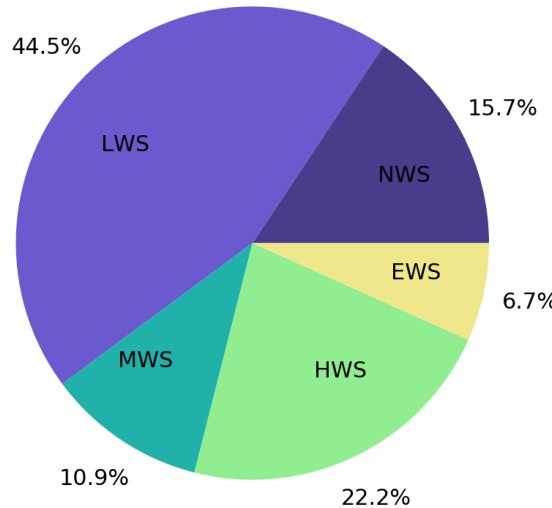

**Figure 8.** Share of time in each shear exponent range.

| NWS | LWS | MWS | HWS | EWS |
|-----|-----|-----|-----|-----|
| $\alpha \leqslant 0$ | $0 < \alpha \leqslant 0.14$ | $0.14 < \alpha \leqslant 0.2$ | $0.2 < \alpha \leqslant 0.4$ | $\alpha > 0.4$ |

**Table 3.** Definition of the ranges of $\alpha$.

## 3.3 Low-level jet statistics

In average, LLJs detected with the method introduced in Sect. 2.3.2 are observed 15.5% of the time during the experiment, their mean wind speed at the core is 9.4 m s$^{-1}$ with a mean core altitude of 168 m. As illustrated in Tab. 2 and in Fig. 9, their occurrence, height and strength are varying with month, wind direction and time of the day.

Figure 9a presents the number of occurrence of LLJs as function of the time of the day, the occurence is the highest during the night, very low between 10 am and 3 pm, and increasing after 3 pm towards the night in a clear diurnal pattern.

Figure 9b provides the distribution of LLJ by wind direction, while the color scale indicates the wind speed of the LLJ core. Most of the LLJs (58.4%) are coming from the [15° - 90°] wind sector corresponding to a *land wind*. A significantly smaller proportion (19.7%) is coming from the north-west direction, blowing along the coastline.

Figure 9c presents the distribution of LLJ core speed (black line) in an histogram and two sub-distributions corresponding to *daytime* and *nighttime* occurrence of LLJs. The *night-time* distribution has significantly more spread than the daytime one and
is shifted towards higher speeds. In the figure, the grey histogram reflects the hub height wind speed distribution as in Fig. 6a for comparison. Note that this histogram has a different vertical scale (LLJ events are detected only in a small portion of all profiles) and that the core of the LLJ is mostly detected higher than the hub explaining the high values. It can be seen that the general distribution is similar, in shape, to the *daytime* LLJ core speed distribution, but the *nighttime* LLJ core speed histogram



WIND
ENERGY
SCIENCE
DISCUSSIONS

**Figure 9.** Statistic of LLJ detection. (a) Occurrence of LLJ events by the time of the day, purple background denoting nighttime. (b) Wind rose of LLJ events. (c) Histogram of the wind speed at the core of all the detected LLJs (black), at night (blue), during the day (orange) compared to the histogram of the wind speed at hub-height (grey). (d) Statistical monthly variability of LLJ core speed. (e) Distribution of LLJ heights over all period (black) at night (blue) and during the day (orange). The hatched area represents the rotor area. (f) Statistical monthly variability of LLJ core height.

is shifted to the right by about 3-4 $\mathrm{m.s}^{-1}$, meaning that wind speeds in LLJ cores are on average higher than those at the hub
height (at least of this reference turbine).



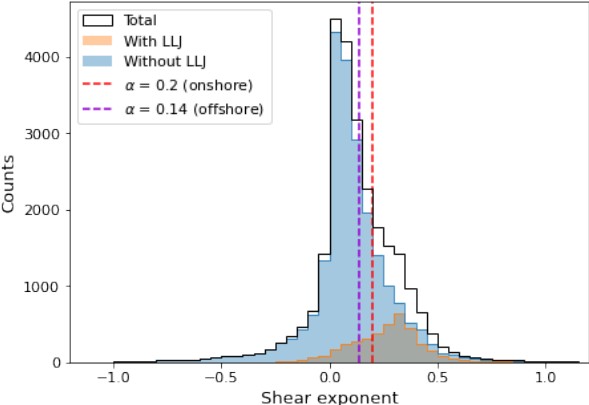

**Figure 10.** Wind shear histogram in LLJ and non-LLJ cases

Figure 9d illustrates the monthly variability of LLJ core speed in the form of a boxplot, with the width of the boxplots proportional to the LLJ occurrence for each month. The monthly mean wind speed is indicated by a grey dashed line for comparison and its spread is illustrated by the two grey areas. Once again, LLJ core speeds are on average higher than hub height speeds, but follow roughly the same monthly trend.

Similarly to Fig. 9c and d, Fig. 9e and f) present statistics on the height of the LLJ detected. In Fig. 9e, histograms of LLJ core height for *daytime* and *nighttime* do not show a significant diurnal variation. LLJ core heights are ranging from 50 to 350 m above the sea level, although most of the observations are between 100 and 250 m. The hatched area in both figures corresponds to the wind turbine rotor, and indicates that most of the LLJ cores are observed in the upper half of the rotor area. The monthly evolution of the LLJ height (Fig. 9f) does not show a very clear trend, it is most of the time well inside the rotor

area. A slight increase of both core speed and core height is nevertheless observed in June, corresponding to the month with the lowest number of LLJ observations. In April and May, LLJs cores are observed at slightly lower altitudes than in the other months of the dataset. Although, these observations are to be analysed with care as the spread of values for each month shown by the boxplots is rather high.

### 3.4 LLJ and high wind shear crossed statistics

Given that a LLJ is by definition a maximum in the wind profile located at a low altitude, its presence often implies high wind shear values from the core down to the ground. LLJs are not the unique responsible for high shear, but the link between these two phenomena can be explored by comparing the number of LLJ observations simultaneously with a shear events as defined in Tab. 3. In Fig. 10, the histogram of the shear exponent for all the data (black), no LLJ (blue) and when LLJ are detected (orange) clearly show that LLJs are associated to higher shear, 0.27 for LLJs compared to 0.12 for all the database and 0.1

for no-LLJ data only. In all the dataset, events with $\alpha > 0.2$ represents 29% of the time, and if only cases without LLJ are considered this ratio goes down to 22%. In contrast, when only accounting for LLJ events, 68% of these are $\alpha > 0.2$.



|  | NWS | LWS | MWS | HWS | EWS |
|---|---|---|---|---|---|
| LLJ | 5.5% | 5.6% | 13.2% | 34.5% | 35.4% |

**Table 4.** Percentage of presence of LLJ in each $\alpha$ range.

## 3.5 Application to a 10MW wind turbine

In this section, the power production of the 10 MW turbine is analysed using the 10-min wind speed at hub height and the turbine's power curve to estimate a 10-min electricity production. This simple approach does not take into account all the complexity of the wind energy harvesting (wind shear across the rotor, the turbulence intensity...) but enables to put into perspective the measurements performed.

Over all the period, the turbine would be operating 81.5% of the time with a capacity factor of 43.9%. This is in agreement with capacity factors usually encountered offshore Costanzo et al. (2023). Taking into account only production time, it would operate at rated power 13.56% of the time and the average 10-min power production is 5410 kW.

In the period, spanning mainly during spring and summer, daytime represents 59% of the total time and nighttime 41%. But the mean nighttime production (6104 kW) being higher than during daytime (4882 kW), 51% of the total electricity is produced during the day and 49% at night. On a complete year, it is likely that nighttime will contribute the most to the total electricity production.

LLJ are representing 15.5% of the overall recording time and 17.6% of the electricity production. However, the mean production during LLJs (5362 kW) is slightly lower than the mean production (5411 kW). Having a production share higher than the share of time while having a lower mean power production might be counter-intuitive. One reason for that is, even if the LLJ wind speed at hub-height is slightly lower than average, LLJs are producing 93% of the time compared to non-LLJ events that are producing only 74.5% of the time. Then, their higher regularity to produce electricity make them globally profitable.

The production time (Fig. 11a) and the total production (Fig. 11b) during LWS and MWS events, in line with the recommendation of IEC (2009), represent, by far, most of the production time (61%) and nearly 63% of the total production. Higher shear exponent (HWS and EWS) are representing 30% of the time and 33% of the electricity production with a mean 10-min production about 15% higher than the mean (Fig. 12b). NWS are representing 9% of the total production time and only slightly more than 4% of the total electricity production. This suggests that NWS are mostly appearing during low wind speed events. The mean energy production during NWS is 55% lower than the mean production (Fig. 12b).

The cumulative sum of the production times, shorted in descending order, depicted in Fig. 12a, shows that the 27.6% best production times are representing half of the total power production and the 50% best production time are representing 79% of the total production. This underlines the fact that most of the energy production is made by a reduced number of high production periods.




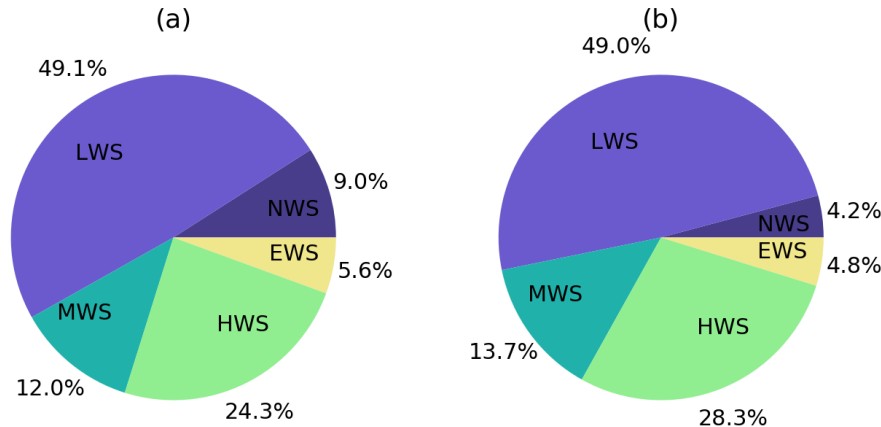

**Figure 11.** Participation of each shear exponent range to (a) the total production time and (b) the total production.

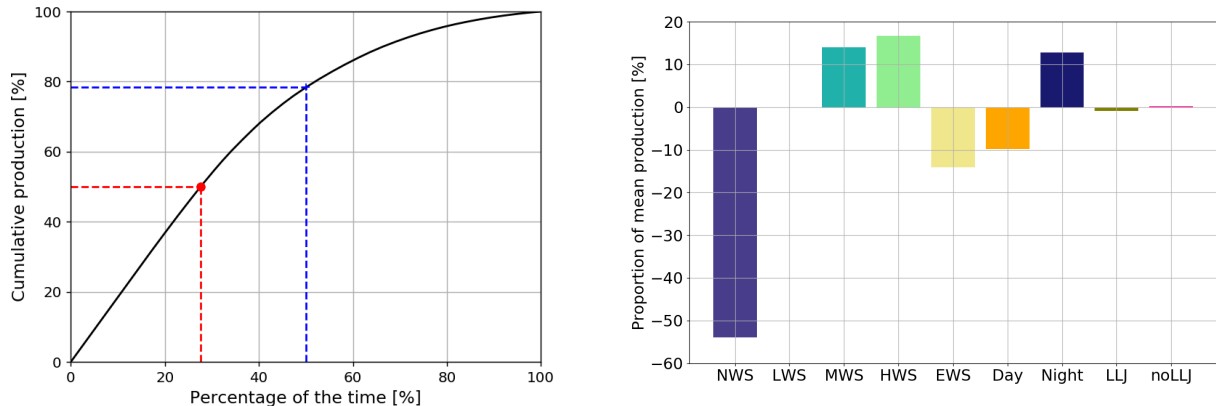

**Figure 12.** (a) Cumulative sum of 10-min production periods sorted in descending order. (b) Difference of mean 10-min production in each category compared to mean 10-min production.

Figure 12b summarises the ratio between the mean power production of each type of event relative to the overall mean 10-min production. It is clear again that the shear exponent is highly influencing the power production, the night/day also, but LLJs as such, are not a determinant factor.

## 4   Discussion

The methodology described in Sect.2 shows how both time and space information accumulated from sparsely spaced vertical
sLiDAR scans can be used to increase the vertical resolution. For that, the use of horizontal homogeneity hypothesis is success-fully validated in the present configuration for both horizontal wind speed and horizontal wind direction with a high correlation





coefficient ($r^2 > 0.99$) regardless of the wind direction. This result is likely to be valid above a homogeneous surface such as the sea but care should be taken closer the shore or in presence of non-homogeneities such as islands. The 10-min statistical convergence analysis (Sect. 2.2.3 assumes that the integral turbulent scale that is proportional to the height, this approach can be

discussed but the approximation is also made for the USA, so it is likely to have no significant consequence in the conclusions. The turbulent scale was also assumed uniform regardless of the direction of the LiDAR with respect to the wind direction. In reality, this is not the case in the atmosphere as turbulent structures are elongated in the stream-wise direction. This approach is also expected to be conservative.

LLJs observed near the coast and described in Sect.3.3 can be the result of multiple physical processes. At the site, LLJs

were mostly observed to come from the land (Northeastern direction), during nighttime, with a monthly rate of observation varying in the range 8-30%. Even if the deployed measurement setup allows for measurements up to 500 m, LLJ core were mostly detected between 100 and 250 m. These observations do not fit to the description of coastal LLJs made by Parish (2000); Nunalee and Basu (2014) or by Rijo et al. (2018) who observed, at several places around the world, synoptic driven coastal jets blowing parallel to the coast and intensified by mesoscale forcing, especially by cross-coast barocline due to temperature

gradients and/or the topography change at the sea/land transition. The range of core height of these coastal LLJs are of the order of the marine boundary-layer (300-500m), which is also much greater than in the present study where heights of 100-250 m are observed. Our observations are probably resulting from other processes. In the study of Svensson et al. (2019), authors observed LLJs near the coast of the Baltic sea blowing normal to the coastline. They suggest that these nighttime LLJs may be formed by warm air advection over the sea or by advection of nighttime LLJs generated from the surrounding land

surfaces toward offshore, crossing the coastline. The latter possibility can fit the observations, where nocturnal LLJs, classically observed over land during nighttime, for example Mitchell et al. (1995), would remain some kilometres offshore. In addition when wind blows from the land, the coastline is the site of a rough to smooth transition likely to favour an acceleration of the wind speed close to the surface. At night, the streamwise surface temperature gradient imposed by the coastline may also play a role. The data available during the present test campaign are too limited to answer these questions and to explain the

condition of persistence of land-generated LLJs further offshore.

The analysis of the power production of a 10 MW offshore wind turbine proposed in Sect. 3.5 is made with the type of turbine really installed in the offshore wind park at some 10 km from the measurement site. The choice of a higher turbine could result in slightly different conclusion regarding power production and shear exponent as the hub would be closer to the LLJs mean height. Globally, the power is expected to be increased, but also the shear exponent. Another limitation of the

analysis is that power production was estimated only from hub-height information, as conventionally done, a more advanced approach would be needed to account for the available power on the wind turbine sweep area.

## 5   Conclusions

This paper presents an original methodology to measure a well-resolved vertical profile of the horizontal wind speed above the sea surface up to 500 m. By configuring a scanning Doppler wind LiDAR to make PPI scans from the shore at only 6



elevation angles above the sea surface, the methodology allows for measuring a vertical profile from 20 m to 450 m above the sea level with 27 intermediate altitudes thanks to a well validated hypothesis of horizontal homogeneity within a limited range. This configuration made it possible to reach altitudes not available with classical profilers and is of major interest for both increasingly tall offshore wind turbines and understanding more globally ABL processes at play for wind energy applications.

Taking into account the spatial resolution of each single PPI scan, the statistical error associated to the convergence of the
10-min horizontal wind speed estimation from the proposed method is found to be in the same order of magnitude of that of a ultrasonic anemometer or a wind vane placed in the same conditions. The error shows a dependence on the angle between the wind direction and the scan and is found slightly lower that a USA for wind speed below 10 m s$^{-1}$ and slightly higher for wind speeds above, indicating good reliability of the approach.

Using a sLiDAR from the shore eliminates the need for a stabilisation and/or compensation for support motion (such as
when installed on a buoy) as a disadvantage, the range of the device, here 3 km limits the distance to the shore where a virtual mast can be reconstructed. A longer range scanning wind LiDAR might be deployed in a similar way to go further away from the shore. A possible further improvement of the method would be to evaluate its capability to estimate the turbulence level.

The validated method is applied to a 7 month test campaign in the French Atlantic coast generating a unique dataset that allows to assess the wind resource and to document the presence of LLJ and high wind shear events never reported in the
North-East Atlantic coast to date. In the observation data, periods with a shear exponent higher than the IEC are frequently observed, more than 30% of the time, and would represent 33% of the total energy production of a 10 MW wind turbine. During these periods, the wind turbine would operate away from its design generating higher loads and, possibly, anticipated fatigue.

During the test campaign, low-level jets are observed about 15% of the time and contributes to 17% of the production of a
potential 10 MW wind turbine installed at the site. Even if they are not part of the strongest production events recorded, it is to be noted that, in the period observed, energy is produced for 93% of the time when LLJs are observed, which is higher than the average production time without LLJs that goes below 80%. Another observation is that LLJs are often associated with HWS and EWS if we consider that 68% of LLJ observations are linked to the across-the-rotor shear coefficient higher than $\alpha = 0.2$, compared to 22% of the time when considering non-LLJs events. These high wind shear conditions well out of the
IEC range can, once again, be very demanding for the structure of the wind turbine and induce fatigue loads not accounted for in the design.

The presence of LLJs well inside the rotor area of a 10 MW wind turbine (mainly in the range 150-200 m) indicates their importance for wind energy. On the one hand, they are beneficial to the total production mainly due to their regularity but, on the other hand, they are associated with a shear exponent well above design value. These observations are key to better
understand the coastal wind resource all the more than the presence of LLJs is often underestimated by meso-scale models (Nunalee and Basu, 2014).

During the test campaign, LLJs are mostly observed when wind is coming from the land (northeastern wind) and during nighttime suggesting that they are generated inland, crossing the coastline, and persisting to reach the observation site located 1.5 km offshore. The origin of the observed LLJs remains uncertain, the punctual profile provided by the sLiDAR being

insufficient to completely verify any hypotheses. This work opens the question of the dynamical evolution of land-generated
        nocturnal LLJs when crossing the coastline and going further offshore, and the role played by the strong roughness and thermal
        land-sea transition is yet to be revealed. More globally, parameters driving the spatial extend of land-sea micro-meteorology
        need to be further investigated to understand how complex processed created at or modified by the coastline can impact the
        wind conditions some tens of kilometre offshore where wind farms operate. This work brings directions to try to understand

and document more precisely the variability of the offshore wind resource close to the coast where the dynamics of complex
        processes remain unrevealed.

*Data availability.*  Raw data of each figure is made publicly available in a data repository (to be done)

*Code and data availability.*  under consideration

*Author contributions.*  B.C. conceived, designed and performed the measurement campaign, A.V. proposed and implemented the data treat-
      ment methodology, A.V and B.C. performed the data analysis, A.V. wrote the basic version of the paper (as her Master thesis), A.V. and B.C.
      made writing—review and editing, B.C. supervised the project and received the funding. All authors have read and agreed to the published
      version of the manuscript.

*Competing interests.*  Author declare that no competing interest is present.


*Acknowledgements.*  The SEM-REV test site (Centrale Nantes) is deeply acknowledged for hosting the field experiment and its technical
support. The scanning LiDAR installation was possible thanks to the framework of the WEAMEC, West Atlantic Marine Energy Community,
with funding from the Pays de la Loire Region and Europe (European Regional Development Fund) under the WAKEFUL project. This work
also contributed to the MATRAC research effort sponsored by ANR-ASTRID under contract ANR-18-ASTR-0002.



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
