# Peer review of "Measurement and analysis of high altitude wind profiles over the sea in a coastal zone using a scanning wind LiDAR - application to wind energy"

_Wind Energy Science, 2023_

## Author Comment (AC1)

**In this answer to reviewers, the original comments of the reviewers are in black font and the answers of the authors in blue.**

RC1: 'Comment on wes-2023-141', Anonymous Referee #1, 15 Nov 2023 reply

**General comments**

This paper analyses 7 months of scanning lidar measurements of winds at multiple heights at a coastal site in northwest France, with the aim of characterising the vertical profile, assessing vertical wind shear and the frequency of low-level jets (LLJs) to understand their impact on wind energy. The manuscript presents relevant and interesting analysis, however the novelty of the study is unclear and the aims and key points could be more clearly defined and emphasised throughout the text to strengthen the narrative. One major concern is that the lidar measurements have not been independently verified. It is recommended that this and several further points below should be considered to improve the manuscript before publication.

We would like to thank the reviewer for the detailed and valuable remarks. In general, we tried to explain the novelty of the work throughout the text, but following your comment we revised it to add some clarity. As for the specific comments, please find our answers below:

**Specific comments**

1.  Measurement reliability, validation/verification: a key issue is that the scanning lidar measurements have not been independently verified against an independent reference source and therefore it is unknown how reliable the measurements are (and therefore that the measurement setup is correct). If there is an absence of any other in-situ reference sources near the site, the study could at least compare the measurements with reanalysis (e.g. ERA5) to show that they agree in terms of the overall wind conditions.

    Following this remark, a scatter plot comparing the ERA5 data at the nearest grid point to the sLiDAR reconstruction data was made and included in the methodology section. It is to be noted that ERA5 reanalysis is not expected to perform well in such a complex coastal area due to its spatial resolution of (0.25°x0.25°) and overall low accuracy compared to direct measurements. In the absence of other "independent" sources of data, the comparison is limited to a rather rough validation of the general trends. This comparison is presented and discussed in section 2.3.

2.  Measurement methodology: the text says that this methodology is original but it is not clear from the text (until the conclusion) how the method presented is actually new and differs from a standard wind field reconstruction of PPI scans observed at different elevation angles.

    Following this remark, the introduction and methodology sections are revised to emphasize originality.

3. Wind profiles: A major aim is to analyse high altitude wind profiles, however no wind profiles (ie. speed vs height) are presented. The manuscript could benefit from showing the mean profiles over the period, and greater insight would be provided by showing these relative to night, day, during LLJ, without LLJ etc, as is partitioned in the other analysis. Analysis of wind veer would also add further insight of relevance to wind energy.

Following this remark, an example of the time/height evolution of the wind speed is depicted in figure 7 (new version) with an example of a profile to illustrate the resolution of the profile. Corresponding text is added in a new section 3.1. Note that averages by time of the day or by direction were not added as it would imply averaging over very different meteorological conditions, which, in our opinion, brings little value to the analysis.

Wind veer would indeed be an interesting analysis. We added this as a perspective, but it is out of scope for the present study.

4. The application section (S3.5) is weak. Wind energy is advancing towards larger turbines and therefore the section should additionally assess a larger turbine(s) (e.g. IEA 15MW or 18MW) given that the measurements presented specifically go to higher heights which are directly relevant for such turbines. The study also has the advantage of the detailed wind profile observations which can be used to derive rotor equivalent wind speed (e.g. https://wes.copernicus.org/articles/5/1169/2020/) which is more appropriate for the analysis, and would provide a more accurate understanding on the impact of shear on power production across the rotor area(s).

Following this remark, section 3.5 was rewritten using the IEA 15MW wind turbine. The rotor equivalent wind speed approach was also implemented to improve the evaluation of the power production. It was chosen to address only one turbine, because several turbines would over-complicate the graphs.

5. Seasonal biasing: The measurements are limited to 7-month period centred on the Summer season and therefore are seasonally biased. Moreover, this part of the year is the period of lowest wind speeds and therefore lowest wind power production. The implications of this should therefore be discussed and caveated.

Following the remark, this limitation is emphasized and discussed at several places in the text including:

sec 3.2

"Availability in July and August below 65\% is due to periods when the sLiDAR was used for other purposes than the test campaign described in this article. As a consequence, statistics for these months should not be taken as fully statistically

representative of the period. Additionally, the total database does not cover a full year. For these two reasons, overall statistics in Tab. 2 and Fig. 8 are not representative of yearly statistics.."

"This pattern is typical for this area, yet, it is should not be considered as an annual statistical representation as all the winter season and a large part of the autumn are not included in the dataset. Western winds may be under-represented in the dataset."

sect 3.6: "It has to be noted again, however, that the dataset does not cover the whole year and that some months are underrepresented, thereby limiting the generalization of a month-to-month statistical analysis."

and "Over a full year, it is likely that night-time will contribute the most to the total electricity production (considering the longer nights during autumn and winter). Additionally, the colder part of the year, which is not included in the observation period, is expected to be more productive due to stronger winds."

6.  Meteorological conditions: the results would benefit from incorporating additional analysis of the meteorological conditions during the measurement period (e.g. temp, pressure, precipitation, humidity, atmospheric stability). This would provide deeper insight in physically interpreting the wind profiles, vertical shear and LLJ events and help backup the physical reasoning made to explain the results observed. If no in-situ measurements are available then reanalysis could be used to examine this.

    In the literature, mostly onshore, it is clear that atmospheric stability often plays a role in explaining LLJ and resulting wind profiles. It is probably also the case here. However, thermal stability evaluation is particularly complex at a coastal site because of the lack of adequate methodology and instrumentation. Most of the common methods cannot be applied (direct flux measurement, Richardson bulk) or would lead to erroneous interpretation due to the internal boundary layer developing at the land/sea transition. In these transitional zones, it seems that only temperature profiles would be exploitable, but such data are not available in the experiment. Note that the determination of offshore thermal stability is one of the great challenges to tackle for a better wind resource assessment, and the coastal zone is an additional complexity. From past experience, relying on reanalysis data (such as ERA5 with a 0.25°x0.25° grid) so close to the land-sea transition is particularly delicate and we think would result in more questions than answers. Numerical simulations at microscale are currently running as part of another research action aiming to achieve better representation of specific LLJ events observed in the present data; they may lead to a deeper understanding of the thermal contribution.

    As for pressure and humidity, their presentation (in time series?) over 7 months would not bring any crucial information in themselves.

7. Introduction wrt low level jet importance: A key aim of the paper is to characterise LLJ frequency but the introduction does not explain clearly how LLJs specifically impact wind power and wind turbine performance and exactly what other studies have found. For example, the term 'quite frequent' describing LLJ frequency from literature (line 48) is extremely vague.

The interest of studying LLJ and high shear events is more clearly emphasized at several locations in the text. Comparison to existing studies are added with quantitative data.

Especially the introduction is modified to: " As LLJs can occur at the operating altitude of an offshore wind turbine, several consequences such as a modification of the load distribution on the rotor, wake recovery rate, and turbulence level are expected to affect the overall performance of the wind turbine and the lifetime of the structures, making the study of their characteristics of significant importance in coastal wind energy projects. To date, offshore and coastal LLJs are studied mainly in the North and Baltic seas, closed seas, and along the northeastern coast of the USA. Their presence in the European Atlantic northwestern coast remains largely unstudied."

8. Section 2:
    1. Horizontal homogeneity: the scanning lidar wind field reconstruction method relies upon the assumption of horizontal flow homogeneity. In Section 2.2.2 it should be more clearly emphasised that the comparison made shows that the wind fields measured by the scanning lidar are relatively homogeneous. By definition, it appears that the NRMSE threshold filter essentially removes heterogenous flow cases which is why the correlations in Section 2.2.2 are so high? This should be commented on. Also, it would be possible to use the data to obtain an estimate of how many horizontally heterogenous vs homogeneous flow events are present in the measurements. This would be a very interesting result to report, especially given the land-sea coastal transition zone, and help further back up the homogeneity assumption.

        During the data processing, wind speed/direction reconstruction from each single PPI scan assumes homogeneity **within the scan** for the fitting procedure, and NRMSE is used to discard cases where the fitting is too poor (can be due to inhomogeneity, but not only). In section 2.2.2, we propose an increase of the vertical resolution using different scans, the hypothesis of horizontal homogeneity is necessary **between two reconstructed points** that can be up to 1 km away. In Fig 4 the increase of the NRMSE threshold in the wind speed/direction reconstruction is made to decrease the uncertainty on wind speed/direction estimation in order to focus on the comparison between points from two scans and evaluate horizontal homogeneity. Giving the set-up the increase of the NRMSE indeed filters out non-homogeneities within each scan but not potential non-homogeneities normal to the coastline

between two points at 1km distance. From the analysis illustrated in Fig 4 we conclude that, despite the coastline, the homogeneity is good.

Evaluation of homogeneity/heterogeneity of the wind near the coast is an interesting topic but analysis based on just two positions seems nevertheless too limited. Using a larger number of points and especially along PPI1 would probably be the most suited approach to evaluate the evolution of wind speed/direction as a function of the distance from the coast, wind direction and thermal conditions. This idea is included in the perspectives.

Sec 2.2 is revised, mostly by adding an introduction to 2.2.1, 2.2.2 and 2.2.3:

"Shimada (2020) demonstrated the validity of the horizontal wind speed and direction reconstruction at a single point from a PPI scan by a comparison with a fixed profiling LiDAR. The approach relies on the homogeneity of the wind within the PPI scan. In this section, we propose to use the same reconstruction method (see Sec.2.2.1) but to extend the methodology in order to have a profile reaching 500m with a high vertical resolution. To reach this goal, several PPI scans are performed within a 10-min period at several heights and a profile reconstruction method is proposed based on horizontal homogeneity and discussed in Sec.2.2.2. In Sec.2.2.3 the consequences of performing 6 PPI scans in 10-min are discussed in terms of statistical convergence."

2. Figure 4: standard practice for validation is to include the results from a linear regression.

   Figure 4 shows the scatter plot of horizontal wind speed at two locations with the same height. A linear fitting is applied, the coefficient is added.

3. The backlash effect on the scan head may impact the pointing accuracy especially when scanning at multiple elevations. Has this been examined and mitigated?

   The hard-target procedure also allowed us to check for eventual backlash in the gears. Alternative clockwise/counterclockwise PPI did not allow for detecting a backlash bigger than 0.05°.

   Text modification: "While pointing at the reference pylon, no backlash bigger than 0.05° was detected in azimuth."

4. Section 2.1: more details should be given about the hard target procedure as different methods exist and Shimada et al., (2020) used a specific approach called 'soft target calibration'. This is important as the viability of the measurement heights relies on this process.

Indeed, the reference to Shimada (2020) to illustrate the hard-target method is misleading as they refer to the method but they don't use it. The reference is removed and more details are given in the text:

"The hard-target procedure was applied using a reference pylon situated at 1.9~km from the LiDAR. The precise measurement of the GPS position of both the pylon and the sLiDAR enabled the determination of the azimuth angle ($\phi$) with an uncertainty of 0.5°. Pitch and roll angles of the sLiDAR were adjusted using internal inclinometers with an uncertainty of 0.1°. The reference pylon was also used to estimate the remaining elevation error.[...]"

5. Line 106 – Please clarify in the text that -29dB is the minimum CNR threshold applied. Normally an upper CNR threshold is also applied to ensure that any high returns from hard objects are removed.

   The sentence is clarified to:

   "In this work, as in Gryning (2019) and in Paskin (2022), the minimum carrier-to-noise ratio (CNR) threshold value used to validate a RWS measurement was set to -29dB. No upper CNR threshold was applied."

6. Does the obtaining u and v from the linear regression of Eqn 3. yield the same results as the Shimada et al., (2020) method? If not, the statements of reported accuracy (119-121) may not hold.

   Yes, the linear regression leads to the same results as Shimada (2020). This is added to the text.

7. Section 2.2.3: Please provide a physical explanation of why the USA exhibits a high sensitivity to wind speed and the scanning lidar a low sensitivity.

   More details are given in the text:

   "Figure 5 (middle) shows that $N_{USA}^{10\,min}$ is very sensitive to wind speed variation. Indeed, the statistical convergence of a single point is only a function of the number of independent measurements acquired in 10 min and with the hypothesis of a time scale independent from the wind speed, the faster the wind speed, the higher the number of points measured in a 10 min time. On the contrary, the sLiDAR can count on spatial measurements, regardless of the wind speed."

- Section 3:
  1. This section would benefit from explaining to the reader more clearly why specific results/statistics are being presented and physically what the results mean.

The choice of the proposed variables and results are described in more detail in the text, including the introduction of section 3.

2. Section 3.1: It is not clear what the percentage frequencies in Table 2 are calculated relative to. Are % reliable profiles defined as number of reliable profiles relative to the full time period? Is the LLJ occurrence a percentage relative to the number of profiles observed or relative to the full time period (i.e. total number of 10-min periods in a month). These definitions need to be made clearer in order to interpret the statistics properly.

Availability is computed relative to the full time period. LLJ occurrences are based on validated 10 min observed profiles. This is specified in the text.

3. The low data availability during July and August is a concern as it brings into question how representative the measurements are of this period and therefore of the LLJ frequency. Please discuss and caveat this.

Sentences at several places are rephrased to emphasize the limitation, also linked to a previous remark.

For example: "Availability in July and August below 65\% is due to periods when the sLiDAR was used for other purposes than the test campaign described in this article. As a consequence, statistics for these months should not be taken as fully statistically representative of the period."

4. Frequencies in histograms might be easier to interpret if y-axis (counts) were expressed as a percentage relative to the observed period.

The main reason why counts were used instead of more conventional percentages is because several histograms are combined in one plot for comparison purposes: the whole dataset is divided into two parts, both parts are represented as histograms, alongside with a separate histogram for the full dataset. Since percentages are normally calculated as the ratio between the number of counts falling within the given bin to the total number of counts in the dataset represented by the histogram, plotting all three histograms separately would make their comparison non-trivial. On the other hand, defining percentages for the smaller histograms relative to the total number of counts in the full dataset would yield the same graph in terms of shapes, but the newly defined percentages would have little statistical sense. Using counts for the vertical counts gives us a good overview of the statistics of the dataset and its subsets and simultaneously allows us to compare them.

5. Section 3.4: would be useful to comment on why LLJ are linked to high shear (stability?).

The question of the link between LLJ and high shear is discussed in section 3.5 (previously 3.4) where it is demonstrated that LLJ are statistically linked to higher wind shear.

The question of the nature of the link (why, how) is more complex to answer with the data we have available, some elements are nevertheless discussed in the discussion section.

**Technical corrections**

Please re-check the grammar throughout, listed below are only some examples:

Line 3 – 'tip' should be 'tips'

Done

Line 21 – 'form' should be 'from'

Done

Line 22 – 'to' – reads better is it's 'from'

Done

Line 24 – Please define the approximate distance range from the coast within which the meteorological land/sea (coastal) transition region occupies and add reference(s) to back this up.

The paragraph was rephrased with a reference to Roger (1995).

Text modification: " From a meteorological point of view, see Roger (1995), the effects of the sea/land transition can still be visible at 100 km offshore, so even with the advent of floating offshore turbines that tend to increase the averaged distance to shore, most of these installations can be considered coastal rather than offshore. "

Line 38 – 'observation' should be 'observations'

done

Line 45 – 'Older and more recent studies' change to 'Previous studies'

done

Lines 45-48 – for readability the list of references would be better moved to a bracketed list at the end of the sentence.

done

Line 51 – references stated wrongly – should be '(Soares et al., 2014; Svenson et al, 2019)

done

Line 55 – what 'consequences' are expected of LLJs on wind energy in terms of wind power generation and mechanical loads? Need to be more specific so that the motivation of studying them is clearly identified.

examples are given

Line 64 – please define what is 'low data availability' from the sLidar in the Wagner study – the statement 'data availability was rather low' is too vague. Also include the frequency of LLJs that this study found

The text is modified to give more details about this study:

" Wagner (2019) used a sLiDAR on the FINO1 platform in the North sea in DBS mode (vertical profiling) to probe up to 518 m. Although data availability was rather low (11.9%) due to time-discontinuous LiDAR measurements and partial instrument failure, they successfully analyzed LLJ events within a one year period and provided tentative explanations of their generation. At the site, LLJs were detected 14.5 % of the time and on 64.8% of the days."

Line 101-102 – rotation direction – this is more clear if changed from 'direct and indirect' to 'clockwise and anticlockwise' (whatever way around is correct).

done

Line 178 – what is meant by the 'cord' of a scan?

The straight distance between the start and the end of a scan at a given range.

Section 2.2.3: lots of equations are given within the text body which makes them difficult to track, and therefore it might be clearer to define equations on separate lines. Define z_alpha, k and z.

Important equations are separated.

z_alpha/2 refers to the confidence level chosen for the statistics. Text modified to: "where z_alpha/2=1.65 refers to 90% confidence level"

**RC2**: ['Comment on wes-2023-141'](), Anonymous Referee #2, 18 Nov 2023

This paper presents an interesting study of the use of a scanning lidar for profiling the wind speed close to the shore. It also presents a climatology of the low level jets and an analysis of the resulting energy yield from a wind turbine. There are some points which should be addressed before publication:

Authors would like to thank the reviewer for the detailed review. Please find our answers to the comments below:

1. The novelty of the study should be emphasised. From the background in the paper, it would seem that this is not the first time a scanning lidar has been used to determine profiles. Also, has there been any previous work to look at the climatology of low level jets in this part of the Atlantic cost?

   Sections of the text were revised to make clearer the novelty of the methodology (multiple heights) and that this is the first study in this part of the Atlantic coast. Introduction and methodology section mainly.

2. As mentioned in this paper, there have been studies of LLJs in neighbouring areas such as the North Sea (e.g. https://wes.copernicus.org/articles/4/193/2019/). A quantification of such studies would be beneficial for comparison with this Atlantic site.

   LLJs at the coast can have multiple origins which makes it complicated to compare observations at different locations. Additionally, a quantitative comparison with multi-year statistics such as in the proposed reference is complex as our dataset contains only a limited number of months (not a full year). Clarifications on this topic are added in the introduction and tentative comparison is presented in the discussion.

   It is highlighted in the introduction that the nearest studies of LLJ were performed in the North sea (nearly closed sea with finite fetch) and US east coast and no study exists in the European Atlantic coast (to the best of the authors' knowledge).

3. The construction of a profile ought to be validated with a profiling lidar, e.g. on a floating platform. Has this been done previously? Is there any scope for validation with the set-up here?

   The reconstruction of the horizontal wind speed/direction over the sea using a PPI scan at a single elevation angle was validated in Shimada (2020) against a co-located profiling LiDAR (both on a fixed platform). The added value of the present work is to use several PPI scans at various elevations and at several ranges within each PPI scan to increase the vertical resolution of the final profile allowing for its fine analysis. Therefore, in the methodology part, the present article focuses on the validity of performing multiple PPI scans at different elevations and on the horizontal homogeneity that allows the vertical reconstruction.

The Methodology section is modified to explain better what needs to be validated and how it was done.

4. The work suggests that there is not much horizontal variation in the wind speed within 500 m of the intended measurement point. Whilst the paper acknowledges that turbulent fluctuations will play a role in the difference between the validation pairs, there is still quite some scatter from the regression lines. At such a relatively short distance from the coast, a developing marine internal boundary layer is likely to affect the correlation between measurements depending on the prevailing stability conditions and wind direction (land/sea, sea/land). The work would benefit from an analysis of how the correlation varies with these parameters. In the case of stability, if actual measurements are not available, this could be inferred from a reanalysis.

At such a short distance from the coast, effects of the coastline are expected (roughness/thermal transitions) but our results suggest that the homogeneity hypothesis still stands at the site. Sec. 2.2.2 "This shows that for this study, and despite the land-sea roughness and thermal transition due to the coast, the evolution of horizontal wind is not significant within a 1000 m horizontal range centered at the virtual met mast, thereby verifying the horizontal homogeneity (for the purposes of this particular study)."

Evaluation of homogeneity/ heterogeneity of the wind near the coast is an interesting topic. A careful analysis would, however, require the use of a larger number of points than those proposed in this study. Analyzing points along PPI1 would probably be a possible approach to evaluate the evolution of wind speed/direction as a function of the distance from the coast, wind direction and thermal conditions. This idea is included in the perspectives.

5. As mentioned by another reviewer, inclusion of some actual profiles (instantaneous and/or mean) would be beneficial, e.g. by time of day, season, direction, LLJ, non-LLJ.

To illustrate the performed measurements, a time series of the vertical profile during a nocturnal low-level jet event is included in the form of a color plot (Fig. 7). A single profile is also extracted to show the vertical profile resolution. Supporting text is added in a new section 3.1. Averages by time of the day or by direction would imply averaging meteorological situations which are, in our opinion, too different from each other to allow for a clear analysis of the profile.

6. Some of the terminology is not consistent, e.g. 'shore wind' in figure 6c rather than 'land wind' (presumably?).

> Text and figures are revised to be consistent with the definition where sea wind and land wind are used to denote winds coming from the sea [135◦-315◦] and from the land [315◦-135◦], respectively.

7. It is stated that the day/night and sea/land wind distributions are very similar. Some explanation is given, but it is acknowledged that no directional analysis was done (line 258). It would seem highly likely that the winds are coming from the land at night due to thermal effects and thus most of the LLJs also will be coming from the land at night. This ought to be a simple analysis to do and should be included.

> This sentence was misleading. Statistically, the day/night filtering does not reveal directional prevalence. Similarities in the distributions are rather linked to roughness and thermal properties, both sea wind and daytime are associated to a low shear and both nighttime and land wind are associated to a high shear.

> The text is modified:

> "In Fig. 7b, where wind shear exponents are sorted into *land wind* and *sea wind* categories, the *sea wind* distribution largely resembles the *daytime* distribution and the *land wind* distribution is very similar to the *nighttime* distribution. However, a day/night filtering does not directly correspond to a directional prevalence. Even though a systematic link between the pairs cannot be made based only on the similarity of the histograms, a connection can be made. On the one hand, *night-time* over land is often associated with a stable thermal stratification that leads to a higher shear exponent compared to unstable conditions often observed during *daytime* and associated with lower shear exponents. On the other hand, *sea wind*, due to lower surface roughness, is associated with a lower power exponent compared to *land wind*. Therefore, both sea wind and daytime are associated with a low shear and both nighttime and land wind are associated with a high shear explaining the observed similarities. Similar observations of the influence of thermal stability and surface roughness on the wind shear are reported in Hanafusa et al. (1986); Irwin(1979)."

> About LLJs (section 3.3), they are clearly linked to land wind at nighttime, but it cannot be deduced that all land winds happen during night-time (LLJs are measured only 15% of the time).

8. Line 239: I assume this should refer to 'sea wind' and land wind'?

> Indeed that is a mistake, the sentence is corrected to:

> l293 "Later in the text, *sea wind* and *land wind* are used to denote winds coming from the sea [135°-315°] and from the land [315°-135°], respectively."

9. In figure 9, the blue boxes, whiskers and circles are not defined or if they are, I didn't fully understand the explanation.

> The boxplots are defined in the text where Figure 9f is introduced (p16).

10. A more recent turbine (larger) would be better to use, e.g. IEA 15MW or even 22MW as these are more representative of the next generation turbines being sited offshore.

    Following this remark, section 3.5 was revised with the IEA 15MW wind turbine.

11. The analysis of the impact of jets would be more enlightening if the profile shape was taken into consideration (e.g. rotor equivalent wind speed). The link to the mean speed is interesting but is quite a limited analysis and not a direct consequent of the jet shape, only its magnitude at hub height which is quite localised (particular to this site) and also dependent on the nine months of data.

    Following this remark, the rotor equivalent wind speed approach was implemented and results and analysis in section 3.5 were re-done accordingly.

12. The English could be improved in places, e.g. the verb 'to allow' does not catenate, i.e. cannot be followed directly by 'to' + infinitive.

    Associated English corrections were made.